# Exploring the Impact of Studying abroad in Hungary on Entrepreneurial Intention among International Students

**Jingjing Wu** *[ID] **and Ildikó Rudnák** [ID]

Doctoral School in Economic and Regional Sciences, Hungarian University of Agriculture and Life Sciences (MATE), 2100 Gödöllő, Hungary; Rudnak.Ildiko@uni-mate.hu
* Correspondence: jingjing.wu.jj@gmail.com

**Abstract:** With the global development of the regional mobility of education, Hungary has gradually become a priority country for overseas students to pursue tertiary education. Meanwhile, the experience of studying abroad can prepare international students to engage in international business and new entrepreneurial activities around the world. In this context, the research on the change of entrepreneurial intention brought about by studying abroad in Hungary deserves great concern and exploration. Given that, this paper contributes to finding out whether the entrepreneurial intention (EI) of international students has changed after coming to Hungary and what environmental factors would bring about changes in entrepreneurial intention before (EI-before) and after (EI-after) coming to Hungary. Here, the environmental factors tested include multiple network construction (MNC), overseas entrepreneurial perception (OEP) and multicultural cognition (MC). Additionally, an Exploratory Factor Analysis, Paired Samples *t*-Test and Hierarchical Multiple Regression Analysis were conducted to process data. The output reveals that after coming to study in Hungary, the entrepreneurial intention (EI-after) of international students has indeed been improved, whilst OEP and MC make a significant impact on the entrepreneurial intention to a similar degree under the control of demographic variables and EI-before.

**Keywords:** entrepreneurial intention; international students in Hungary; multiple network construction; overseas entrepreneurial perception; multicultural cognation



## 1. Introduction

Entrepreneurship is generally regarded as the engine of innovation and social growth [1], and entrepreneurial activities become a channel for knowledge spillover to transform into economic growth and competitiveness [2]. The development of entrepreneurship is not only the need of the contemporary economy but also one of the key points of economic development in all countries [3]. The related benefits of entrepreneurship are reflected at both the individual and social levels, such as self-employment, improved living standards, higher life satisfaction, poverty reduction, as well as enhanced social cohesion and well-being [4–6]. People who pursue entrepreneurship are more likely to experience higher job fit because their personal values are consistent with the work characteristics of this career path [7]. Furthermore, Dzisi and Odoom [8] believe that economic growth may be limited without the participation of adventurers, due to entrepreneurship being the catalyst for economic development. In fact, the development of entrepreneurship and the change in people's attitude towards entrepreneurship is a challenge faced by almost all countries [9]. That is because different environments may lead to different social realities, there are differences in the factors that make entrepreneurship feasible [1]. The opportunities for individual discovery and exploitation may also be affected by numerous factors [10].

With the continuous development of internationalization, regional mobility has undoubtedly been greatly promoted, which has led to the success of global higher education and increased the possibility of career development of international students [11]. According to Education at a Glance 2020 [12], the international enrollment of higher education

in the world reached 5.6 million in 2018, of which 3.9 million were attracted by OECD countries and 1.7 million by non-OECD countries, and the number is increasing year by year. In this circumstance, Helms et al. [13] mention that the experience of studying, living and working abroad can prepare for a future career in international business and new entrepreneurship around the world. That is because cross-cultural experience enables people to enter a knowledge environment that is completely different from that of their own countries and contribute to acquiring advanced knowledge, skills and new ideas, thereby enhancing their ability to identify entrepreneurial opportunities [14–16]. Moreover, even a short stay for foreign educational exchanges has also been shown to affect a person's ability to identify profitable business opportunities [16]. Meanwhile, previous studies have pointed out that overseas experience would increase the knowledge stock of returnees and increases their "social capital" [17,18]. In sum, overseas experience has a positive impact on the possibility of engaging in entrepreneurial activities [2,19,20].

For Hungary, the Erasmus+ and Stipendium Hungaricum scholarship programs have accelerated the influx of international students. The Tempus Public Foundation presents that the number of foreign students receiving higher education in Hungary has been on the rise, and the number has increased 3.5-fold from 11,783 students in 2011 to 38,422 students in the 2019/2020 academic year [21]. Given that, international students in Hungary have become an integral part that should not be ignored by higher educational institutions. Likewise, it is conjectured that the experience of studying in Hungary is likely to enhance the entrepreneurial intention of international students as well. Nevertheless, the existing literature mainly studies the entrepreneurship of local Hungarian students or compares the entrepreneurship of Hungarian students with that of other countries [3,22–24]. Little or no attention has been paid to the research on the entrepreneurial intention of international students in Hungary.

Additionally, the study on the entrepreneurial intentions of international students is a global research topic, but it has not received sufficient attention. In view of the small number of previous studies, the research mainly explores the existing entrepreneurial intentions of international students and possible predictors [25,26]. However, the change of intention brought about by the experience of studying abroad has not been excavated. At this point, our study attempts to address this persisting gap identified in the previous literature and add new relevant lines in this domain. First, entrepreneurship is a social activity, which has different needs according to the specific context, and the study of entrepreneurial intention urgently needs to warrant attention to the contextual and temporal aspects [27]. Thereby, this study is conducted from the perspective of international students in Hungary, which enlarge the yet uninvestigated group of people and fills the gap in the existing entrepreneurial intentions mainly for local Hungarian students. Second, this study provides a theoretical relationship between the changes in entrepreneurial intention and the specific environmental factors brought about by studying overseas in Hungary. Understanding students' specific entrepreneurial intentions will help relevant institutions clarify the factors that can be considered in cultivating entrepreneurial intentions among them [28]. Therefore, an in-depth understanding of international students in Hungary can undoubtedly help relevant institutions to take appropriate measures to sustainably strengthen and promote the emergence of entrepreneurial behavior.

In this study, data were collected from international students in Hungary by an electronic questionnaire, and the SPSS software was used to analyze the data. The results show that the entrepreneurial intention of international students has indeed changed after coming to study in Hungary and the entrepreneurial desire has been improved. What is more, two environmental factors (OEP and MC) have an impact on the formation of entrepreneurial intention after controlling for the effect of demographic variables and entrepreneurial intention (EI-before). However, the proportion of their impact is not high, whilst the result reflects that multiple networks construction (MNC) has no significant impact on entrepreneurial intention (EI-after), which is unexpected. In this case, the corresponding measures need to be implemented to strengthen the extent of these impacts.

The structure of this paper is as follows. After this introduction, the next section introduces the theoretical background and the hypotheses to be tested. Section 3 expounds on the materials and methodology for empirical analysis. Then, Sections 4 and 5 present the main research results and their discussion. Lastly, Section 6 includes conclusions, suggestions and limitations.

## 2. Theoretical Background and Hypotheses

### 2.1. Entrepreneurial Intention

The key to understanding the entrepreneurial process is having entrepreneurial intention and being seen as the first step in a long and complex entrepreneurial process [29]. The study of entrepreneurial intention has existed in the entrepreneurial literature for decades. The generally accepted study by scholars began in the 1980s, when Shapero [30]'s entrepreneurial event model was put forward. Afterwards, Ajzen [31]'s theory of planned behavior become another predominant view entering the field of vision. Shapero [30]'s model emphasizes the phenomenon of entrepreneurial event, which is influenced by the perception of desirability (personal value system and social system to which the individual belongs) and the perception of feasibility (financial support and potential partners). The decision to start an entrepreneurial activity requires a pre-existing belief that the activity is desirable and feasible, as well as an individual's tendency to take action on opportunities and certain types of triggers. Ajzen [31]'s model is based on individual intention, which is the result of three determinants: attitude towards behavior (personal evaluation), subjective norms (social pressure) and perceived behavior control (ability to implement behavior). It is the basis for understanding the relationship among attitude, intention and behavior and focuses on how the cultural and social environment affects human behavior. Both models regard intention as a predictor of entrepreneurial behavior [32,33]. That is, the perception of feasibility is consistent with the behavior control of perception, and the perception of desirability aligns with the attitude towards behavior [34]. Krueger et al. [33] further support these two models. Intention is the single best predictor, while individual variables and situational variables contribute only a small amount of explanatory power to entrepreneurial behavior.

In order to determine the level of entrepreneurial activity, it is very useful to understand, study and investigate entrepreneurial intentions, which can provide valuable insights and help [9]. The psychological structure related to entrepreneurial intention, self-efficacy, personality characteristics, risk-taking tendency and personal initiative is regarded as some of the variables most related to entrepreneurial behavior [5,35]. Moreover, Shah et al. [9] indicate that previous studies have found that students' entrepreneurial intention has a strong correlation with demographic factors and entrepreneurship education. When individuals have more knowledge and understanding of entrepreneurship as a professional profession through courses and training, perceived personal knowledge and skills may induce their intention to put their knowledge into practice [5,36,37]. However, there are significant differences in the perception of the desirability and feasibility of entrepreneurial behavior among different countries and different genders [38]. Students in developing countries think that entrepreneurship contributes more to society and their evaluation of entrepreneurship is higher than that of their peers in developed countries [39]. Regardless of their entrepreneurial intentions, women show lower self-esteem and lack of confidence in their entrepreneurial success [32,38].

Entrepreneurial intention is affected not only by personality traits but also by environmental factors [40]. When society supports entrepreneurship, individuals are more likely to make this choice because they feel that the environment around them approves their decision to become entrepreneurs [36]. Such as political and economic factors, social background and perception of opportunities and resources [5]. In addition, the dynamic characteristics of job insecurity, increased demand for services and high unemployment in the labor market make individuals begin to consider non-traditional employment such as self-employment [35]. With regard to social capital, individuals are affected by the

valuation of their more intimate environment, which may be related to their closer ties with family or friends [36]. Dabic et al. [32] find that family support for entrepreneurial activities has a positive impact on entrepreneurial intention, and it is even more decisive for women who are willing to start a business.

### 2.2. Foreign Experience and Entrepreneurship

With the continuous development of the internationalization of the company and the globalization of the labor force, global competence has become an important ability to discover business opportunities [41]. Entrepreneurs are engaged in the business of discovering and taking advantage of opportunities, so entrepreneurs with a global perspective are more likely to take advantage of the resources unique to the international market [13]. Therefore, international mobility is regarded as an opportunity for organizational work (corporate expatriation) and entrepreneurship (expatriate entrepreneurship) [35]. Helms et al. [13] conclude that, for students, the experience of studying, living and working abroad can help students engage in international business and new entrepreneurship around the world for future careers. Sommer expresses that the existing international academic education experience can well overcome the obstacles caused by the formation of international entrepreneurial intention [42] and has a positive impact on the risk of transition to entrepreneurship [2].

### 2.2.1. Multiple Network Construction

Schröder et al. [43] research confirms the importance of having a personal network related to successful entrepreneurship. Due to the unfamiliarity of the environment, many student entrepreneurs urgently need to get in touch with new people and rely on existing possible connections [44]. Prior studies have pointed out that overseas experience not only increases the knowledge stock of returnees, but also increases their "social capital" [17]. Being in a foreign country gives returnees access to professional networks and sources of advanced knowledge and new ideas [14]. Returnees may be able to maintain social relations in the host country after returning to their home countries, which enables them to continue to update their technology. This social capital helps them still have access to different sources of information and knowledge when they return to their home countries [2]. However, academic entrepreneurship of international students forms another dimension for entrepreneurial mobility, but there is a lack of evidence that cooperation with entrepreneurs who return after graduation or after completing postdoctoral tasks may benefit the universities of the host country [45].

### 2.2.2. Overseas Entrepreneurial Perception

Lai and Vonortas [2]'s research show that returnees are more likely to engage in entrepreneurial activities. Studying or working abroad enables people to enter a completely different knowledge environment from their own country, providing them with an opportunity to acquire advanced knowledge and new ideas [15]. Likewise, research by Pinto [46] points out that participation in the Erasmus project has a causal and positive impact on becoming an entrepreneur because the university graduates' international mobility helps students master foreign languages quickly and they are positioned by the market as entrepreneurial mobile employees. Moreover, local governments in some Chinese cities, such as Beijing, Shanghai and Zhejiang, have also implemented a series of policies to attract graduates studying abroad to return home according to their career preferences. For example, providing generous subsidies and venture capital for students abroad enables them to contribute to the economy of their hometown [47]. In addition, higher education institutions have a responsibility to train and prepare students to work in a dynamic, fast-changing entrepreneurial and global environment. Introducing International Entrepreneurship Education to students can help students start a business abroad or start exporting [8].

### 2.2.3. Multicultural Cognition

Personal factors play a vital role in the entrepreneurial process of returnees, which is largely determined by the direct experience of studying abroad, as well as the cultural, social and economic environment of the host country [2]. Harris et al. [48] and Alon et al. [49] find that the factors that affect entrepreneurial intention vary from culture to culture and national characteristics and cultural attitudes are important factors. Pinto [46] examines the impact of Spanish mobility on labor market outcomes and skill development, then concludes that the experience of students studying abroad has a positive impact on the possibility of becoming entrepreneurs, working abroad and improving information communication and communication skills to the foreigner. Meanwhile, the cultural background and language skills of returnees enable them to make use of non-local experience and knowledge [14]. Compared with their local counterparts, the companies founded by returnees are more innovative. When returnees start new businesses, their multicultural knowledge, overseas education and work experience may be another important driver of innovation [15].

Based on the above theoretical literature review, the hypotheses to be tested are as follows:

**Hypothesis 1 (H1).** *There is a significant difference in entrepreneurial intentions of international students between before and after coming to study in Hungary.*

**Hypothesis 2 (H2).** *The multiple network construction of international students in Hungary has a great impact on their entrepreneurial intention.*

**Hypothesis 3 (H3).** *The overseas entrepreneurial perception of international students in Hungary has a great impact on their entrepreneurial intention.*

**Hypothesis 4 (H4).** *The multicultural cognition of international students in Hungary has a great impact on their entrepreneurial intention.*

## 3. Materials and Methods

### 3.1. Theoretical Framework and Measure

This study aims to test whether the experience of studying in Hungary affect the entrepreneurial intention of international students, that is, whether their entrepreneurial intention has been changed after coming to Hungary for studying. Here, the entrepreneurial intention before coming to Hungary (EI-before) will be compared with that after coming to Hungary (EI-after). Moreover, after coming to Hungary, international students will be impacted by the new environment to a certain extent, which will lead to a change of their mindset. Therefore, what are the specific environmental factors that affect the change of entrepreneurial intention of international students? Moreover, to what extent is it affected? In respect to this, the potential environmental factors explored in this study include multiple network construction (MNC), overseas entrepreneurial perception (OEP) and multicultural cognition (MC). A reference to the model of the study is shown in Figure 1, which includes a temporal dimension and a theoretical dimension. The dimension of time measures the discrepancies in entrepreneurial intention between before (EI-before) and after (EI-after) coming to study in Hungary. The theoretical dimension measures which specific environmental factors (MNC, OEP and MC) affect their entrepreneurial intention after coming to Hungary (EI-after).

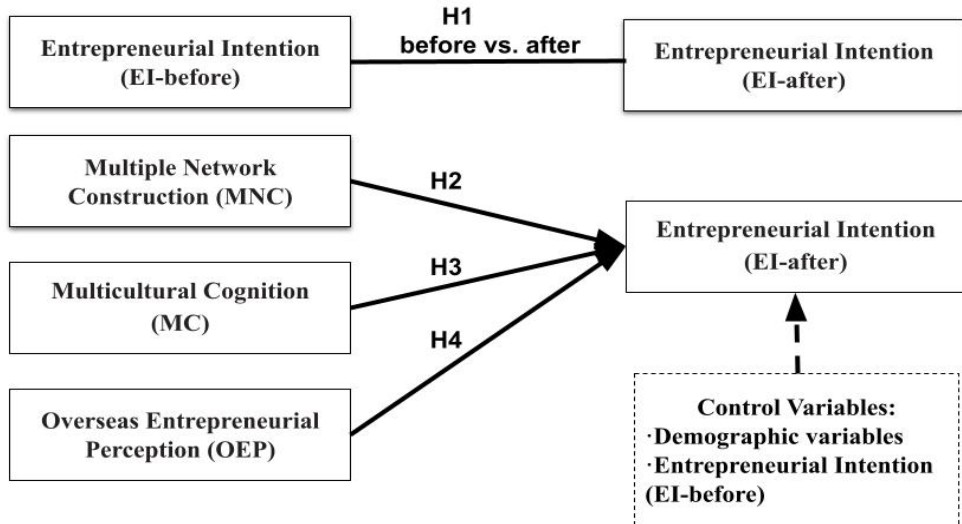

**Figure 1.** The model of theoretical framework.

Meanwhile, Table 1 shows the specific items asked in the questionnaire, in which entrepreneurial intention includes 6 items. The items from EI2 to EI5 are referred to the entrepreneurial intention scale invented by Thompson [50]. In addition, in order to fit the research context, EI1 and EI6 have been added. Moreover, it is worth noting that after comparing the entrepreneurial intentions before and after coming to Hungary, the data of entrepreneurial intention after coming to Hungary (EI-after) are used to analyze the impact of environmental factors and are regarded as a dependent variable. The time period after coming to Hungary is defined as arriving in Hungary until the time point of data collection.

**Table 1.** Specific items for the measurement of variables.

| Dependent Variable | Items |
|---|---|
| Entrepreneurial Intention (EI) (After coming to Hungary) | EI1. I have a sense of entrepreneurship<br>EI2. I plan to start a company in the future.<br>EI3. I have been looking for entrepreneurial projects and opportunities.<br>EI4. I spend time learning entrepreneurial knowledge and other people's entrepreneurial experience.<br>EI5. I have saved money or considered the source of funds to start a company.<br>EI6. I hope to get wealth and a sense of achievement through starting a business. |
| **Independent Variables** | **Items** |
| Multiple Network Construction (MNC) | MNC1. I have established contact with Hungarian universities.<br>MNC2. I have established contact with a business partner in Hungary.<br>MNC3. I have established contact with potential clients in Hungary.<br>MNC4. I have established contact with relevant enterprises in Hungary.<br>MNC5. I have established contact with investors in Hungary. |

**Table 1.** *Cont.*

| | |
|---|---|
| Overseas Entrepreneurial Perception (OEP) | OEP1. The experience of studying in Hungary has expanded my entrepreneurial horizons and possibilities. OEP2. Hungarian universities promote and encourage students to start a business, resulting in a strong entrepreneurial atmosphere. OEP3. The experience of studying in Hungary made me find the opportunity to start a business. OEP4. Hungarian universities provide students with education, resources and policy support for entrepreneurship. OEP5. The background of studying abroad helps me to get preferential policies or financial support for entrepreneurship when returning home country. OEP6. Studying in Hungary has enhanced my foreign language skills needed for starting a business. |
| Multicultural Cognition (MC) | MC1. I am well aware of the differences between the culture of my own country and that of Hungary. MC2. I can quickly adapt to Hungarian culture and life. MC3. I can understand and adjust the conflicts brought about by multi-culturalism. MC4. I know how to communicate with Hungarians and students of different nationalities. MC5. I am very interested in the culture and customs of Hungarian and students with different cultural backgrounds and often have cultural exchanges with them. MC6. I can change my behavior and cognition according to different cultural needs. |

Here, multiple network construction (5 items) includes contacts between international students and Hungarian organizations or individuals. Then, overseas entrepreneurial perception (6 items) is the education, knowledge, resources, opportunities and so on perceived after coming to Hungary. Multicultural cognition (6 items) is the ability to discover and adapt to the cultural differences brought about by Hungarian and other foreign students. These three factors are considered as independent variables and from the summary of studying experience in Hungary and related literature.

In addition, Entrepreneurship research finds that demographic variables are strong predictors of entrepreneurial intention. Thereby, when detecting environmental factors, demographic variables are set as the control variable. They include gender, age, educational program, studying years in Hungary, working years in Hungary, entrepreneurial experience and family business background [5,32,36,38,39,51,52]. Meanwhile, in order to more accurately detect the impact of entrepreneurial intention caused by environmental factors of studying abroad in Hungary, the existing entrepreneurial intention of international students before coming to Hungary (EI-before) will also become a controlled variable and cannot be ignored.

*3.2. Data Collection, Sample and Analytical Method*

The survey data were from international students currently studying in Hungary and were distributed in the form of electronic questionnaires. Online forms were filled

out anonymously by students. At the beginning of the questionnaire, the students were informed of the purpose of the study and obtained informed consent. Moreover, the main period for the collection of data was from March 2021 to May 2021. A total of 500 questionnaires were sent out, from which 345 were fully answered and involved in this study. The respondents mainly included international students from Hungarian University of Agriculture and Life Sciences, Corvinus University of Budapest, Budapest University of Technology and Economics, Eötvös Loránd University and Budapest Business School-University of Applied Science, which are located in the Hungarian capital of Budapest and surrounding cities. Among them, 100 questionnaires were distributed to each university. In regard to the questionnaire, it is divided into three sections. The first section is about the demographic characteristics of the participants. In the second section, participants are required to fill in the scale of their entrepreneurial intention (EI) before and after coming to Hungary, which is used to detect whether their entrepreneurial intention has been changed. After that, the third section is to detect the potential environmental factors that affect the change caused by coming to study in Hungary, including multiple network construction (MNC), overseas entrepreneurial perception (OEP) and multicultural cognition (MC). The specific items questioned in sections two and three are listed in Table 1. In addition, a five-point Likert scale is used to allow participants to show the extent of their agreement for all items, ranging from 1 = strongly disagree to 5 = strongly agree.

The following are the demographic profiles of participants displayed in Table 2. The share of gender was similar, with 165 (47.8%) males and 180 (52.2%) females. Most of them were between 25 and 29 years old (44.3%), followed by those under 24 years old (30.4%) and older than 30 years old (25.2%). Educational programs included exchange students (2.6%), Bachelor's (22.6%), Master's (49.6%) and PhD (25.2%) programs. The proportion of respondents who studied in Hungary for less than 1 year and 2–3 years was similar at 40% and 43.5%, respectively, while those who had studied for more than 4 years accounted for relatively few, 16.5%. Furthermore, more than half of the respondents had no work experience in Hungary (59.1%), while about one-third of respondents had less than one year's work experience (31.3%). In addition, 73.9% of the respondents had no entrepreneurial experience, and 79.1% of them had no family business background.

Therewith, collected data were processed using SPSS 26.0 software and was mainly arranged into three parts. First of all, exploratory factor analysis (principal component analysis, Varimax rotation) was conducted to explore the relationship between items among the dependent variable and independent variables, so as to refine and reduce items to form a small number of related subscales [53]. Subsequently, in order to analyze whether the entrepreneurial intention of the participants has changed before and after coming to Hungary, the paired-samples *t*-test tool in SPSS was used to test hypothesis 1. What is more, to have a clearer understanding of which specific items have been changed, the six items in the entrepreneurial intention (EI) are all compared separately. In the last part, hierarchical multiple regression analysis was used to test, after controlling for demographic variables and EI-before, whether multiple network construction (MNC), overseas entrepreneurial perception (OEP) and multicultural cognition (MC) have an impact on entrepreneurial intention after coming to Hungary. Multiple regression is used to explore the relationship between a continuous dependent variable and many independent variables or predictors, which can provide information about the entire model and the relative contribution of each variable that makes up the model [53]. Therefore, it is suitable for data analysis in this part to obtain outcomes.

**Table 2.** Demographic profiles of sample.

| Demographic Variables | Item and Code | Frequency | Percent |
|---|---|---|---|
| Gender | Male = 1 | 165 | 7.8 |
| | Female = 2 | 180 | 52.2 |
| Age | Less than 24 years old = 1 | 105 | 30.4 |
| | 25–29 years old = 2 | 153 | 44.3 |
| | More than 30 years old = 3 | 87 | 25.2 |
| Educational programs | Exchange students = 1 | 9 | 2.6 |
| | Bachelor = 2 | 78 | 22.6 |
| | Master = 3 | 171 | 49.6 |
| | PhD = 4 | 87 | 25.2 |
| Studying years in Hungary | Less than 1 year = 1 | 138 | 40 |
| | 2–3 years = 2 | 150 | 43.5 |
| | More than 4 years = 3 | 57 | 16.5 |
| Working years in Hungary | No experience = 1 | 204 | 59.1 |
| | Less than 1 year = 2 | 108 | 31.3 |
| | 2–3 years = 3 | 30 | 8.7 |
| | More than 4 years = 4 | 3 | 0.9 |
| Entrepreneurial experience | No experience = 1 | 255 | 73.9 |
| | Less than 1 year = 2 | 45 | 13.0 |
| | 2–3 years = 3 | 18 | 5.2 |
| | More than 4 years = 4 | 27 | 7.8 |
| Family business background | Yes = 1 | 72 | 20.9 |
| | No = 2 | 273 | 79.1 |

## 4. Results

### 4.1. Reliability

Before analyzing the data deeply, Cronbach's alpha test is used to check the reliability. As shown in Table 3, the Cronbach's alpha reliability coefficient of entrepreneurial intention before and after coming to Hungary is 0.895 and 0.902, respectively. Moreover, the value of Cronbach's alpha of MNC (0.874), OEP (0.845) and MC (0.862) are also presented. Devellis [54] suggests that the Cronbach alpha coefficient of a scale greater than 0.7 is considered to be of high reliability. This means that the data of this study seem to have acceptable internal reliability and are expected to produce pragmatic results.

**Table 3.** Cronbach's alpha reliability of variables.

| Variables | | Cronbach's Alpha | Cronbach's Alpha Based on Standardized Items | N of Items |
|---|---|---|---|---|
| Entrepreneurial Intention (EI) | Before coming to Hungary | 0.895 | 0.895 | 6 |
| | After coming to Hungary | 0.902 | 0.902 | 6 |
| Multiple Network Construction (MNC) | | 0.874 | 0.878 | 5 |
| Overseas Entrepreneurial Perception (OEP) | | 0.845 | 0.845 | 6 |
| Multicultural Cognition (MC) | | 0.862 | 0.864 | 6 |

### 4.2. Exploratory Factor Analysis

To determine whether the dataset is suitable for the application of a factor analysis, Bartlett's test of sphericity and Kaiser–Meyer–Olkin (KMO) are used to judge the adequacy and suitability of the sample. Bartlett's test of sphericity should be significant ($p < 0.05$) and the KMO index should be 0.6 as the minimum value to satisfy the factor analysis [55]. Here, the data are detected to have a KMO value of 0.8, exceeding the minimum value of

0.6, and the results of Bartlett's test are significant ($p = 0.000$). Therefore, a factor analysis is appropriate.

The principal component analysis (Table 4) finds that there are four components with eigenvalues greater than 1, which explained a total of 66.2% of the variances, contributing 33.6%, 16.9%, 8.4% and 7.3%, respectively. Afterwards, Communalities provide information on how much variance is explained in each item. A low value (less than 0.3) could indicate that the item does not match other items in its component [53]. Here, the communality values in Table 4 show that no item is less than 0.3, so all the items are retained and loaded significantly to their component. Moreover, the Varimax rotation technique is performed and showed that the main loadings on Component 1 are items belonging to entrepreneurial intention (EI), and the main loading on Components 2, 3 and 4 are items belonging to multiple network construction (MNC), overseas entrepreneurial perception (OEP) and multicultural cognition (MC), respectively. In addition, there is a reasonable correlation from 0.050 to 0.378 among these four components. Therefore, the results of factor analysis support the use of the EI items, MNC items, OEP items and MC items as separate scales.

**Table 4.** Rotated component matrix.

| Items | Component | | | | Communalities |
|---|---|---|---|---|---|
| | 1 | 2 | 3 | 4 | |
| EI1 | 0.720 | | | | 0.623 |
| EI2 | 0.785 | | | | 0.702 |
| EI3 | 0.850 | | | | 0.801 |
| EI4 | 0.778 | | | | 0.683 |
| EI5 | 0.693 | | | | 0.585 |
| EI6 | 0.775 | | | | 0.691 |
| MNC1 | | 0.441 | | | 0.415 |
| MNC2 | | 0.840 | | | 0.755 |
| MNC3 | | 0.902 | | | 0.840 |
| MNC4 | | 0.838 | | | 0.769 |
| MNC5 | | 0.878 | | | 0.809 |
| OEP1 | | | 0.602 | | 0.563 |
| OEP2 | | | 0.806 | | 0.779 |
| OEP3 | | | 0.600 | | 0.684 |
| OEP4 | | | 0.846 | | 0.768 |
| OEP5 | | | 0.662 | | 0.606 |
| OEP6 | | | 0.552 | | 0.482 |
| MC1 | | | | 0.696 | 0.530 |
| MC2 | | | | 0.721 | 0.604 |
| MC3 | | | | 0.785 | 0.723 |
| MC4 | | | | 0.789 | 0.668 |
| MC5 | | | | 0.721 | 0.596 |
| MC6 | | | | 0.737 | 0.565 |
| Total variance explained | 0.336 | 0.084 | 0.073 | 0.169 | |
| Eigenvalues | 7.773 | 1.941 | 1.673 | 3.893 | |

Note: 1. Extraction Method: Principal Component Analysis; Rotation Method: Varimax with Kaiser Normalization. 2. EI: Entrepreneurial intention; MNC: Multiple network construction; OEP: Overseas entrepreneurial perception; MC: Multicultural cognition.

### 4.3. Paired Samples t-Test

Here, in order to find out whether the entrepreneurial intention of international students has changed before and after coming to Hungary, a paired samples *t*-test is applied. The paired sample *t*-test is suitable for collecting data from one group of people on two different occasions or under two different conditions [53]. Table 5 presents that all six items among the entrepreneurial intention (EI) are compared. The results show that except for the *p*-value of EI2 (I plan to start a company in the future) being more than 0.05, the *p*-values of the other five items are all less than 0.05. Moreover, due to Sig. (2-tailed) reaching a 5% mean significance level ($p < 0.05$) [53], therefore EI1, EI3, EI4, EI5 and EI6

are significant ($p < 0.05$). Subsequently, we judge that there is a significant difference in entrepreneurial intention of international students before and after coming to study in Hungary due to most of the items reaching this standard.

**Table 5.** Paired Samples *t*-Test for entrepreneurial intentions before and after coming to Hungary.

| | Comparison of Entrepreneurial Intention before and after Coming to Hungary | | Mean | Paired Mean Differences | Std. Deviation | Sig. (2-Tailed) |
|---|---|---|---|---|---|---|
| Pair 1 | EI1 | Before<br>After | 3.17<br>3.31 | 0.14 | 1.219<br>1.162 | 0.006 |
| Pair 2 | EI2 | Before<br>After | 3.39<br>3.43 | 0.04 | 1.239<br>1.207 | 0.468 |
| Pair 3 | EI3 | Before<br>After | 3.02<br>3.18 | 0.16 | 1.313<br>1.215 | 0.006 |
| Pair 4 | EI4 | Before<br>After | 2.94<br>3.12 | 0.18 | 1.240<br>1.248 | 0.001 |
| Pair 5 | EI5 | Before<br>After | 2.67<br>2.81 | 0.14 | 1.204<br>1.190 | 0.011 |
| Pair 6 | EI6 | Before<br>After | 3.45<br>3.56 | 0.11 | 1.217<br>1.152 | 0.048 |

Note: EI: Entrepreneurial intention.

Specifically, comparing the mean values of six items in entrepreneurial intention, the results show that the mean values of EI6, EI2 and EI1 are relatively high. It means that most international students hope to gain wealth and a sense of achievement by starting a business. They plan to start a business in the future and have entrepreneurial spirits. What is more, EI4 and EI3 in entrepreneurial intention changed the most after coming to Hungary, with paired mean differences of 0.18 and 0.16, respectively. That is, "I have been looking for entrepreneurial projects and opportunities, and I spend time learning entrepreneurial knowledge and other people's entrepreneurial experience". Therefore, we could conclude that after coming to study in Hungary, international students have taken more substantial actions for entrepreneurship. However, the intention of international students to start a company in the future has not changed much after going abroad. This could mean that having the initial willingness to start a business is also an important and independent determinant for international students to start a business in the future.

### 4.4. Hierarchical Multiple Regression Analysis

In respect of obtaining more rigorous results, hierarchical multiple regression analysis is performed to detect whether multiple network construction (MNC), overseas entrepreneurial perception (OEP) and multicultural cognition (MC) have an impact on the entrepreneurial intention (EI-after) of international students after coming to study in Hungary, as well as their influencing extent under the control of demographic variables (gender, age, educational program, studying years in Hungary, working years in Hungary, entrepreneurial experience and family business background) and EI-before.

Before implementing the multiple regression analysis, several assumptions need to be tested. Tabachnick and Fidell [55] give a formula for sample calculation: $N > 50 + 8m$ (where m = number of independent variables). Here, the study has three independent variables and eight control variables, so the sample size needs 138. The sample size of this study (345) well meets this assumption. Then, the Pearson correlation matrix (Table 6) visually displays the correlation coefficient between each variable ranging from −0.003 to 0.642. When the correlation coefficient between two variables is greater than 0.7, the variable will have the problem of multicollinearity [53]. In addition, the tolerance variance of all independent and control variables are from 0.613 to 0.890 and their VIF are from

1.124 to 1.631, which meet the limitation of no multicollinearity (Tolerance > 0.10, VIF < 10). Therefore, the data meet the assumption of no multicollinearity. Moreover, Normal P-P Plot presents a reasonably straight diagonal line from bottom left to top right. The Scatterplot reveals that the residuals are roughly rectangularly distributed and not more than 3.3 or less than −3.3 [53,55]. In this case, these results ascertain the assumption of normality, linearity and no outlier.

**Table 6.** Pearson correlation matrix.

| Variables | 1 | 2 | 3 | 4 | 5 | 6 | 7 | 8 | 9 | 10 | 11 | 12 |
|---|---|---|---|---|---|---|---|---|---|---|---|---|
| 1. EI-After | 1 | | | | | | | | | | | |
| 2. EI-Before | 0.642 *** | 1 | | | | | | | | | | |
| 3. Gender | −0.003 | −0.056 | 1 | | | | | | | | | |
| 4. Age | 0.110 * | 0.116 * | −0.231 *** | 1 | | | | | | | | |
| 5. Educational program | 0.006 | −0.097 * | −0.170 ** | 0.549 *** | 1 | | | | | | | |
| 6. Studying years in Hungary | 0.219 *** | 0.044 | −0.071 | −0.007 | −0.059 | 1 | | | | | | |
| 7. Working years in Hungary | −0.005 | −0.081 | −0.095 * | 0.018 | 0.092 * | 0.333 *** | 1 | | | | | |
| 8. Entrepreneurial experience | 0.232 *** | 0.375 *** | −0.099 * | 0.281 *** | 0.131 ** | −0.085 | 0.018 | 1 | | | | |
| 9. Family business background | −0.094 * | −0.184 *** | −0.020 | −0.151 ** | −0.074 | 0.041 | 0.103 * | −0.253 *** | 1 | | | |
| 10. MNC | 0.385 *** | 0.434 *** | 0.004 | 0.112 * | 0.004 | 0.177 *** | 0.158 ** | 0.170 *** | 0.003 | 1 | | |
| 11. OEP | 0.500 *** | 0.328 *** | 0.188 *** | 0.063 | 0.007 | −0.009 | −0.025 | 0.152 ** | −0.030 | 0.470 *** | 1 | |
| 12. MC | 0.438 *** | 0.223 *** | 0.004 | 0.080 | 0.138 ** | 0.141 ** | 0.077 | 0.034 | 0.049 | 0.088 | 0.318 *** | 1 |

N = 345, *** $p \leq 0.001$; ** $p \leq 0.01$; * $p \leq 0.05$. Note: EI: Entrepreneurial intention; MNC: Multiple network construction; OEP: Overseas entrepreneurial perception; MC: Multicultural cognition.

After testing the assumptions, the main models of hierarchical multiple regression analysis are evaluated. As can be observed in Table 7, model 1 is a basic model that includes only control variables (demographic variables and EI-before) and model 2 contains all the variables (demographic variables, EI-before, MNC, OEP and MC). The R Square value in model 1 shows that the demographic variables and EI-before explain 46.0 per cent of the variance. After the independent variables have been included, model 2 as a whole explains 58.9 per cent. Furthermore, the R square change value is 0.129 ($p \leq 0.001$), which means that the independent variables explain an additional 12.9 per cent of the variance in entrepreneurial intention (EI-after) after the effects of the demographic variables and EI-before are statistically controlled for. This is an acceptable result.

**Table 7.** Regression analysis.

| Variables | DV: Entrepreneurial Intention (EI-After) | |
|---|---|---|
| **Control Variables** | **Model 1** | **Model 2** |
| Gender | 0.063 | −0.013 |
| Age | 0.002 | −0.005 |
| Educational Programme | 0.094 | 0.037 |
| Studying Years in Hungary | 0.209 * | 0.185 * |
| Working Years in Hungary | −0.028 | −0.040 |
| Entrepreneurial Experience | 0.009 | 0.009 |
| Family Business Background | 0.029 | −0.005 |
| Entrepreneurial Intention (EI-Before) | 0.645 * | 0.494 * |
| **Independent Variables** | | |
| Multiple Network Construction (MNC) | | −0.005 |
| Overseas Entrepreneurial Perception (OEP) | | 0.214 * |
| Multicultural Cognition (MC) | | 0.274 * |
| $R^2$ | 0.460 | 0.589 |
| Overall F | 35.716 * | 43.376 * |
| $R^2$ change | 0.460 * | 0.129 * |

* $p \leq 0.001$. Note: DV: Dependent variables.

For specifically evaluating how well each of the variables relate, model 2 in Table 7 exhibits that the years of studying in Hungary (β = 0.185, $p \leq 0.001$) and EI-before (β = 0.494, $p \leq 0.001$) belonging to control variables have a significant impact on the entrepreneurial intention (EI-after) of international students. The independent variables of OEP and MC make a unique statistically significant impact on EI-after ($p \leq 0.001$), while MNC does not have an impact on it (β = −0.005, $p > 0.05$). In detail, multicultural cognition (MC)

has a greater impact ($\beta$ = 0.274), followed by overseas entrepreneurial perception (OEP) ($\beta$ = 0.214).

In sum, the hypothesis is determined according to the above analysis results shown in Table 8.

**Table 8.** Hypothesis compliance.

| Hypotheses | Sig. | Result | The Test Used in Our Study |
|---|---|---|---|
| H1 | $p$ (EI1, EI3, EI4, EI5 and EI6) < 0.05 | Confirmed | *t*-test |
| H2 | $\beta$ = −0.005, $p$ > 0.05 | Not confirmed | *t*-test |
| H3 | $\beta$ = 0.274, $p$ < 0.05 | Confirmed | *t*-test |
| H4 | $\beta$ = 0.214, $p$ < 0.05 | Confirmed | *t*-test |

## 5. Discussion

This study examines whether the experience of studying in Hungary has an impact on entrepreneurial intention (EI-after) and the potential environmental factors checked include multiple network construction (MNC), overseas entrepreneurial perception (OEP) and multicultural cognition (MC). The paired samples *t*-test in the SPSS software has detected that only the *p*-value of EI2 is greater than 0.05, and the *p*-values of the rest of EI1, EI3, EI4 and EI5 met the 5% significance level between EI-before and EI-after. As five of the six items in entrepreneurial intention have undergone significant changes after coming to study in Hungary, we consider that there is a great difference in entrepreneurial intentions of international students before and after coming to Hungary, and H1 is confirmed. Likewise, Mao and Ye [56]'s research on returned Chinese international students also shows that studying abroad has an essential influence on their entrepreneurial intention. This is because the existing international academic education experience can well overcome the obstacles to the formation of international entrepreneurial intention [42]. Through a specific analysis of the items for entrepreneurial intention, we notice that the mean value of all items is near three, which infers that most of the international students studying in Hungary prefer self-employment. Likewise, Tehseen and Haider [28] believe that entrepreneurship has always been regarded as an attractive career choice for students. As previously explained, the intention that "I hope to gain wealth and a sense of achievement through entrepreneurship" and "I plan to start a business in the future" are the most recognized by international students. However, the idea of starting a business in the future has not changed before and after coming to study in Hungary, indicating that existing entrepreneurial plans also play a prominent role in the entrepreneurial awareness of foreign students. Furthermore, the most obvious change in entrepreneurial intention after coming to Hungary is to take practical actions for it, that is, "I spend time learning entrepreneurial knowledge and other people's entrepreneurial experience", and "I have been looking for entrepreneurial projects and opportunities". Liu et al. [15] agree that studying or working abroad makes people enter a completely different knowledge environment from their home country and provides them with opportunities to acquire advanced knowledge and new ideas.

Afterwards, a hierarchical multiple regression analysis explores the relationship between entrepreneurial intention (EI-after) and potential environmental factors after coming to study in Hungary under the control for the influence of demographic variables and entrepreneurial intention (EI-Before). The result reflects that MC and OEP meet the criteria of significant level ($p$ < 0.05), thus, H3 and H4 are confirmed. Meanwhile, the *p*-value of MNC does not meet the same significance level ($p$ > 0.05), and H2 is not confirmed. In addition, for personal characteristics, only the years of studying in Hungary ($\beta$ = 0.185, $p$ < 0.05) uniquely affect international students' entrepreneurial intentions (EI-after). This conclusion is in line with the view that Keat et al. [40] has emphasized, which is that entrepreneurial intention is affected not only by personality traits but also by environmental factors. The personality characteristics of entrepreneurs are regarded as the assistance or support to

their behavior, but the determinants of entrepreneurs' behavior are determined by the surrounding environment [57]. However, in addition to controlling personal characteristics, entrepreneurial intention (EI-before) is a great influential variable that cannot be ignored ($\beta = 0.494$, $p < 0.05$).

Given that, although two environmental factors (MC and OEP) have an impact on international students' entrepreneurial intention (EI-after), the proportion is not high (12.9%). More specifically, multicultural cognition (MC) has a greater impact ($\beta = 0.274$). Multicultural cognitive ability is regarded as a priority for foreign students to strengthen because this can not only quickly alleviate cultural conflicts, but also accelerate the integration of foreign students into the new environment. Especially before the establishment of multi-networks, understanding the cultural background of the host country is more conducive to promoting entrepreneurial cooperation. As Harris et al. [48] point out, the factors that affect entrepreneurial intention vary from culture to culture, and national characteristics and cultural attitudes are important factors that affect entrepreneurial intention. In addition, the understanding of cultural background can promote the exchange of entrepreneurial information and effective information communication. Pinto [46]'s research also shows that the experience of studying abroad has a positive impact on becoming an entrepreneur, working abroad and improving the ability of information communicate with foreigners. Meanwhile, overseas entrepreneurial perception (OEP) has a similar influence on entrepreneurial intention (EI-after) ($\beta = 0.214$). It mainly focuses on the entrepreneurial policies, education, knowledge, opportunities and atmosphere perceived by international students after coming to study in Hungary. When society supports entrepreneurship, individuals are more likely to make such a choice because they feel that the environment around them approves their decision to become an entrepreneur, such as political and economic factors, as well as the perception of opportunities and resources [5,36]. Ozaralli and Rivenburgh [58] also propose that experience, education, as well as economic and political environment before entrepreneurial behavior are important determinants of entrepreneurship. In addition, entrepreneurs can also further improve their business performance by expanding their insight into entrepreneurial activities and obtaining adequate government support [59]. Furthermore, the result pertaining to multiple network construction (MNC) denotes that there is no impact on the change of foreign students' entrepreneurial intention (EI-after) ($\beta = -0.005$). It can be inferred that the impact of establishing relations with relevant organizations or individuals in Hungary on the formation of entrepreneurial intention is not ideal. From previous research, Multiple network construction (MNC) can be regarded as "social capital" that overseas experience brings to international students. Returnees may maintain social relations in the host country after returning, which enables them to continue to update their technology [2,17]. In addition, establishing links with the host country and becoming trading partners is most likely to promote the formation of entrepreneurship. In this sense, the international students in Hungary have not been fully awakened by this factor and need to be improved.

## 6. Conclusions, Suggestions and Limitations

Entrepreneurship is generally regarded as an important means of national innovation and economic growth. Different social environments and personal factors may produce different entrepreneurial intentions and behaviors. This study takes the international students in Hungary as the research object. They need to quickly adapt to the new environment and change their mindset in the face of various cultural differences. This kind of cultural gap would accelerate their cross-cultural communication and understanding ability. In addition, studying abroad helps them to perceive the trade gap between their own country and Hungary, which enables them to identify entrepreneurial opportunities and generate entrepreneurial intention. This study presents the changes in the entrepreneurial intention of international students and the specific potential environmental factors that affect the changes after coming to Hungary. The results basically meet expectations, the entrepreneurial intention of foreign students is strengthened, and the two environmental

factors have some impact on the formation of their entrepreneurial intention, whereas the proportion of the impact is not high. Furthermore, multiple network construction did not achieve the desired results. In order to strengthen the influence of this factor, the Hungarian government and higher education institutions must take corresponding measures to improve it. Effective guidance to entrepreneurship can speed up the formation of multiple network construction, so as to improve the entrepreneurial intention of international students.

As Donaldson et al. [27] mention, taking into account the expected differences in social location configuration among prospective forerunners, entrepreneurial support must correspond to the current development stage of an individual. Therefore, it is very important to propose corresponding measures according to the current context of international students in Hungary. Some suggestions are as follows. Firstly, the Hungarian government and tertiary education institutions need to attach importance to the entrepreneurial intention of international students and promulgate policies to encourage them to cooperate with Hungarian organizations or individuals, so as to further promote the foreign trade cooperation between Hungary and the third countries. Secondly, most international students in Hungary are taught in English and do not understand Hungarian, so it is critical to creating a platform suitable for international students to obtain entrepreneurial information. Thirdly, due to the different cultural backgrounds of foreign students, providing targeted entrepreneurship-related education and training can stimulate the formation of their entrepreneurial awareness. Lastly, this paper finds out that the construction of multiple networks currently plays no significant role in the entrepreneurial intention (EI-after) of international students, so it needs to be paid more attention and action to enhance the impact. In this sense, it is necessary to provide entrepreneurial practice opportunities for foreign students, organize the network construction of entrepreneurs and create a good entrepreneurial atmosphere.

Consequently, this study is an extension of the previous literature on entrepreneurial intention. It targets international students in Hungary, which expands the population that has not yet been surveyed and fills the gap in existing entrepreneurship research mainly for local Hungarian students. Furthermore, this study provides a theoretical relationship between the change of foreign students' entrepreneurial intention and specific environmental factors. It is emphasized that the experience of studying in Hungary could enhance the entrepreneurial intention of international students, which should be highly considered and encouraged. Necessary support may enhance international students to develop trade activities and economic ties with Hungarian institutions or individuals to a certain extent. Meanwhile, the study also hopes to arouse relevant institutions in third countries to strengthen their attention and policy support to the entrepreneurial behavior of overseas students returning home. In this sense, the study has important implications for entrepreneurs and those who contribute to promoting entrepreneurship, such as educators, support organizations and policymakers. Overall, it is a very meaningful and practical topic to study the entrepreneurial intention of international students in Hungary.

There are some limitations to this study. The geographical location of the research in this paper is mainly concentrated in the Hungarian capital Budapest and its surrounding cities. Therefore, the survey cannot ensure the geographical coverage and sufficient quantity of international students in major Hungarian cities. Furthermore, the proportion of the impact of the three environmental factors on entrepreneurial intention is not high, we will continue to explore other potential environmental factors that could cause the impact. In addition, it should be noted that the entrepreneurial intention of international students is also influenced by the home country to a certain extent. Then, in addition to the influence of the environment of studying abroad, the cultural, social and economic environment of their home country also needs to be further studied.

**Author Contributions:** Conceptualization, J.W.; methodology, J.W.; software, J.W.; validation, J.W. and I.R.; formal analysis, J.W.; investigation, J.W. and I.R.; resources, J.W. and I.R.; data curation, J.W.; writing—original draft preparation, J.W.; writing—review and editing, J.W. and I.R.; visualization, J.W.; supervision, I.R.; project administration, J.W. and I.R. All authors have read and agreed to the published version of the manuscript.

**Funding:** This research received no external funding.

**Institutional Review Board Statement:** Not applicable.

**Informed Consent Statement:** Informed consent was obtained from all subjects involved in the study.

**Data Availability Statement:** The data presented in this study are available on request from the corresponding author.

**Conflicts of Interest:** The authors declare no conflict of interest.

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
