# Peer review of "Exploring the Impact of Studying abroad in Hungary on Entrepreneurial Intention among International Students"

_sustainability, doi:10.3390/su13179545_

Round 1
Reviewer 1 Report
The article is a statistical analysis of some data. In order to present a clear study:
1. It would be adequate if you increase the number of contacted students (more than 345 questionnaires) or at least write which part of the international students currently studying in Hungary is a sum of 500 questionnaires.
2. It wolud be important if you clarify the reasons why you chose the number of 500 questionnaires instead of other number.
3. At Line 241 you say that "The items mainly refer to the entrepreneurial intention scale invented by Thompson". It would be better if you clarify to what extent do the items refer or not refer to that scale.
4. As for the Limitations of the study, you correctly refer to personal characteristics as an important factor you will study in the future but it would be good to clarify whether you think it is possible to study the influence of personal characteristics separately from environmental factors
Reviewer 2 Report
The paper aims to analyse the impact that studying abroad has on the entrepreneurial intention of international students.
My general comments include the following:
Please consider shortening the abstract and provide just the necessary information in this section.
The introduction section is well structured but additional information should be provided on the element of paper novelty; both for the theory and practice.
In the material and methods section please provide the descriptive statistics of your sample or add those as annexes at the end of the paper.
Also, please explain why the authors choose regression analysis and not logistic models, as they analyse entrepreneurial intention.
It is not quite clear how before and after following Hungarian courses have been assessed in terms of entrepreneurial intentions. It has been used in only one construct? And Why? (It would be more proper a follow up survey ....just a thought)
Best,
The reviewer
Reviewer 3 Report
Dear Authors,
Thank you for the opportunity to review your manuscript. You research is certainly of interest, and I would like to make some comments on how to make improvements to provide a more specific contribution to the global body of knowledge.
I provide my review in four different areas: 1) research problem definition, 2) conceptualisation, 3) methods and analysis and 4) conclusions and implications.
1) research problem definition: on page 2, lines 64-71, you provide a justification for the research topic, but this seems to be lacking a logical explanation. What does the study of EI have to do with during and post-study work rights? I think this may be interesting, if either post-study migration is linked with entrepreneurial activities, or if the reputational gain for the Hungarian government in third countries is linked with the entrepreneurial activities of there returning students. However, evidence for this is not provided, nor discussed. Therefore, I suggest providing specific evidence for the impact of post-study home country entrepreneurial activities on international business activities between country of study and home country, and a justification for why EI needs to be studied in this relation.
A better understanding regarding research problems in the EI domain open for research can be gained from systematic reviews of literature, such as Linan and Fayole (2015). Furthermore, Donaldson et al. (2021) calls for more context specific research in EI, which would be a great argument for justifying the topic of research in this paper. Bear in mind though, that this context specificity needs to be explained clearly to the current case, to make the research and its findings specifically applicable to the unique situation of international students in Hungary, and their professional lives afterwards.
Liñán, F., Fayolle, A. A systematic literature review on entrepreneurial intentions: citation, thematic analyses, and research agenda. Int Entrep Manag J 11, 907–933 (2015). https://doi.org/10.1007/s11365-015-0356-5
Donaldson C, Liñán F, Alegre J. Entrepreneurial Intentions: Moving the Field Forwards. The Journal of Entrepreneurship. 2021;30(1):30-55. doi:10.1177/0971355720974801
2) conceptualisation: I believe that your conceptual model requires some refinement. In Figure 1, you show a temporal dimension and a theoretical dimension of the study. The temporal dimension (before and after) and a conceptual dimension (EI, MNC, OEP, MC). In your explanation of the figure, you state that the purpose is to explore the change of EI ‘caused’ by the experiences in Hungary. This suggests to me, that you measure EI BEFORE, and AFTER the Hungarian study experience, while MNC, OEP and MC are measured at the end of the Hungarian study experience. Table 1 suggests that EI is a dependent variable, implying that pre-study data is not considered in the research. So, I suggest clarifying the conceptual model displayed in Figure 1, to match the actual research model, and explain better how the IVs are linked with the DV(s), and perhaps to each other. I could imagine, that perhaps the relationship between the DV and the IVs is not just fully linear. Recognising this possibility to be incorporated in the theoretical model may also allow for greater contribution to the body of knowledge.
I would also like to make a comment on the article title, which seems to be somewhat misaligned with the conceptual model. There seems to be little reference made to the sustainability of EI in the model, however, the concept of the article is more aligned with international EI, considering the purpose stated for studying the EI of international students in Hungary. I suggest to adjust the title to more accurately reflect the article content.
3) methods and analysis: the study conducts a cross-sectional survey. The target population is international students currently studying in Hungary. EI information is requested before and after, making this a retrospective data collection. I presume by ‘after’, the authors either mean at the time of data collection, or at the time anticipated by the students for after the time they finished their studies. This needs to be clarified.
Tables 3 and 4 provide details of the factor analysis. I recommend removing Table 3, and just provide a description of details in a short paragraph. I also recommend presenting in Table 4 only the customary data (please see for example APA standards). The current version seems to be a copy-paste from the SPSS output, which is not how it is usually presented.
The authors collected length of studies in Hungary, degree type, age, etc., and subsequently test for the magnitude of change before and after EI. In my view, it would be prudent to control for the length of exposure, study level, gender and age when assessing the change in EI. The model needs to consider more underlying factors, to avoid erroneous conclusions. May I also clarify why 5% significance levels were chosen for a means difference test?
The way regressions are presented also have close resemblance to SPSS outcomes. I recommend following academic standards as opposed to just copying SPSS output. In particular, I think Table 9 has an error in it. Is VIP supposed to be VIF.
4) conclusions and implications: having confirmed / rejected hypotheses, I think it is important to clearly articulate the contributions of the research to theory and recommendations derived for practice. Currently, these are not clearly made, and linked with the results. It would also help if the article had a clearly defined research question, to which this section could present a theoretically and practically relevant response.
Good luck with your revisions!
Regards,
A reviewer.
Round 2
Reviewer 2 Report
Thank you for responding to all my previous comments.
Best,
Author Response
Thank you for your comment.
Reviewer 3 Report
Dear Authors,
Thank you for attending to my points of review.
I would like to continue our dialogue on the points I have raised in my initial review:
Point 1: Research problem definition
Response 1: Yes, we agree with you and we have seriously considered your suggestion. And we haven’t found that specific evidence about the relationship between foreign student entrepreneurship and Hungarian reputation that you mentioned. Therefore, we decided to delete this content and convert the purpose of the study to fill the gap in the literature and the specific environmental factors brought about by studying in Hungary. The specific contents are presented at from line 71 to 91and from line 553 to 567.
In addition, thank you very much for the article recommended to us, and we citated it from 83 to 86 and from 534 to 536.
FURTHER REVIEW COMMENTS:
You have revised the gap in a much more specific way, and I think that is great! You say that the EI of international students in Hungary has not received scholarly attention, which is accurate. However, EI of international students has been studied elsewhere, and you seem to ignore this literature. I strongly suggest that you incorporate into your argument, that it is a studied phenomenon globally, but not in Hungary, and articulate the reasons why it is fruitful to study this for Hungary, and what we expect to learn from this study in comparison to other studies conducted in other countries.
I have conducted a quick search for articles and found the following papers to look at for international reference:
- https://journal-jger.springeropen.com/articles/10.1186/s40497-018-0136-0
- https://innovation-entrepreneurship.springeropen.com/articles/10.1186/s13731-017-0066-z
- https://www.scirp.org/journal/paperinformation.aspx?paperid=75622
- https://www.eajournals.org/wp-content/uploads/An-Empirical-Study-on-the-Influencing-Entrepreneurial-Intention-Factors-of-International-Students-Based-on-the-Theory-of-Planned-Behavior.pdf
A hook could be a host country focus as opposed to a home country focus, which seems to characterise the international papers, but that is just a possible idea for you to consider.
Point 2: Conceptualisation
Response 2: Thank you for those valuable suggestions. We followed your suggestion and clarified the research model. This content is added to the article from lines 245 to 249 and from line 256 to 261. And the article title has been changed as well. In order to clarify “before and after”, we changed the content to before/after coming to Hungary for studying.
FURTHER REVIEW COMMENTS:
Thank you for explaining this more clearly. I still do not see how the model presented in Figure 1 shows this. Can you please redraw this image, to reflect EI after as DV, EI before, MNC, OEP and MC as IVs? There are plenty of examples in literature to show how to properly visualise these types of models.
Point 3: Methods and analysis
Response 3.1: Yes, ‘after’ has been clarified in the paper from line 256 to 261.
FURTHER REVIEW COMMENTS: ok, thank you, this is fine.
Response 3.2: That's a good suggestion. That Table has been deleted and the contents is presented from line 342 to 344. The table for factor analysis has changed as well, please check Table 5.
FURTHER REVIEW COMMENTS: that’s good, just please double check table numbers, as Table 4 seems to have gotten lost/missing.
Response 3.3: Thank you very much for your valuable advice. After careful consideration, we decided to add personal factors to the analysis. That is, the personal characteristics are set as control variables to analyse the influencing factors brought by environmental factors. In addition, please check Hierarchical Multiple Regression with new Table 8. Moreover, from line 370 to 371, we added a citation to express the 5% significance level for T-Test.
FURTHER REVIEW COMMENTS: this is now much more clear. I just need to query one thing: you did not seem to have added EI (before) to the model, and label EI (after) as simply EI. I think this still remains to give reason for some misinterpretation. Can you please appropriately label EI as EI (after)? Also, even though you talk about the impact of foreign study experiences on EI, you do not control in your model for EI (before). I think this links back to the lack of clarity of the model in Figure 1. If you have the data, and if you want to maintain the argument that EI grows as an impact of international study experiences, I strongly suggest that you include EI (before) in your model as control. Let us hope, that this does not change the oputcome!
Point 4: Conclusions and implications
Response 4: Yes, we agree with you and added the relevant contents from line 75 to 89 and from line 553 to 567.
FURTHER REVIEW COMMENTS: Great! Now, when you state the results of the hypothesis testing, can you please make reference to the specific test result from which you derive the outcome, and give test outcomes as well? I suggest that this is done in a table format.
OVERALL COMMENTS: I need to note, that the English expression of the article now needs some further work. The style and language expression used in the new parts of the paper falls behind the original article, and I encourage the authors to have a look at this as well.
Good luck with your revisions.
Regards,
A reviewer
Author Response
Thank you so much for your comments.
Please see the attachment
